# Host Genetic Background Influences BCG-Induced Antibodies Cross-Reactive to SARS-CoV-2 Spike Protein

**DOI:** 10.3390/vaccines12030242

**Published:** 2024-02-26

**Authors:** Aubrey G. Specht, Melanie Ginese, Sherry L. Kurtz, Karen L. Elkins, Harrison Specht, Gillian Beamer

**Affiliations:** 1Department of Infectious Disease and Global Health, Cummings School of Veterinary Medicine, Tufts University, North Grafton, MA 01536, USA; aubrey.g.specht@gmail.com (A.G.S.); melanie.ginese@tufts.edu (M.G.); 2Center for Biologics Evaluation and Research, Food and Drug Administration, Silver Spring, MD 20993, USA; sherry.kurtz@fda.hhs.gov (S.L.K.); karen.elkins@fda.hhs.gov (K.L.E.); 3Department of Bioengineering and Barnett Institute, Northeastern University, Boston, MA 02115, USA; hms89@cornell.edu; 4Texas Biomedical Research Institute, San Antonio, TX 78227, USA

**Keywords:** BCG, SARS-CoV-2, diversity outbred, genetic background, cross reactive antibodies, heterologous immunity, humoral immunity

## Abstract

*Mycobacterium bovis* Bacillus Calmette-Guérin (BCG) protects against childhood tuberculosis; and unlike most vaccines, BCG broadly impacts immunity to other pathogens and even some cancers. Early in the COVID-19 pandemic, epidemiological studies identified a protective association between BCG vaccination and outcomes of SARS-CoV-2, but the associations in later studies were inconsistent. We sought possible reasons and noticed the study populations often lived in the same country. Since individuals from the same regions can share common ancestors, we hypothesized that genetic background could influence associations between BCG and SARS-CoV-2. To explore this hypothesis in a controlled environment, we performed a pilot study using Diversity Outbred mice. First, we identified amino acid sequences shared by BCG and SARS-CoV-2 spike protein. Next, we tested for IgG reactive to spike protein from BCG-vaccinated mice. Sera from some, but not all, BCG-vaccinated Diversity Outbred mice contained higher levels of IgG cross-reactive to SARS-CoV-2 spike protein than sera from BCG-vaccinated C57BL/6J inbred mice and unvaccinated mice. Although larger experimental studies are needed to obtain mechanistic insight, these findings suggest that genetic background may be an important variable contributing to different associations observed in human randomized clinical trials evaluating BCG vaccination on SARS-CoV-2 and COVID-19.

## 1. Introduction

Bacillus Calmette-Guérin (BCG) is a live attenuated strain of *Mycobacterium bovis* used to vaccinate 130 million children in 154 countries each year against tuberculosis [1]. However, BCG’s impact extends beyond TB. Starting in 1927, studies showed that BCG confers immunity and cross-protection against other pathogens, including viruses [2,3]. And, during the early months of the COVID-19 pandemic, beneficial effects were associated with BCG vaccination; for example, one epidemiological study found that a 10% increase in BCG index (an estimate for a country’s BCG vaccination use) was associated with a 10.4% decrease in COVID-19 mortality [4]. However, a review [5] found that as the pandemic advanced, declined, and receded, additional studies identified positive associations, no associations, and very rarely identified negative associations. The intriguing associations between BCG vaccination and immune responses to SARS-CoV-2 or between BCG and COVID-19 disease outcomes prompted several randomized clinical trials summarized in Table 1. Notably, in the phase III trial ACTIVATE (NCT03296423), BCG revaccination of older patients (>65 years) reduced the risk for COVID-19 in the 6 months following vaccination by 65% [5]. Another study examined BCG revaccination on the immune response generated by SARS-CoV-2 mRNA vaccination and found increased serum cytokines and higher neutralizing antibody titers in the group receiving BCG than the group not revaccinated prior to their SARS-CoV-2 vaccine, suggesting synergy [6]. Additional trials are underway, with results not yet available [7,8].

The inconsistent associations between BCG and SARS-CoV-2 are often attributed to demographics, environment, behavior, or methodological differences [5,9,10,11,12,13,14,15,16,17]; this was also recently reviewed and put into perspective by Noble et al. [18]. Occasionally, differences in association are attributed to BCG’s heterologous effects on trained immunity or acquired immunity [1]. It is possible, but not often discussed, that genetic background may be a contributing variable in BCG’s associations and “off-target” beneficial effects; however, pathogen-specific immunity [19], antibody responses [20], and responses to BCG vaccination [21] are influenced by genetic background or by specific genes and alleles. Further, in the context of epidemiological studies and clinical trial results, the different associations could be appearing because of genetics, since individuals who live in the same geographic region (and therefore are available for recruitment into a given study) are more likely to share common ancestors and immune response alleles than individuals who live in different regions of the globe [22,23]. These observations support the hypothesis of a genetic basis for BCG-induced SARS-CoV-2 immune responses and/or COVID-19 disease outcomes.

To infect host cells, SARS-CoV-2’s surface spike glycoprotein binds the angiotensin-converting enzyme 2 receptors on host cells, [24] and this interaction is blocked by neutralizing antibodies [25], demonstrating a role for B-cell derived (humoral) immunity against SARS-CoV-2. In silico analysis found six hundred and ninety BCG B-cell epitopes homologous to SARS-CoV-2 B-cell epitopes [26], suggesting BCG vaccination could, in theory, generate antibodies cross-reactive to SARS-CoV-2 spike protein. Using immunohistochemistry, another study showed antibody cross-reactivity between SARS-CoV-2 envelope proteins and the mycobacterial protein LytR C protein, which is expressed by *M. bovis* BCG [27], confirming cross-reactivity. Interestingly, in vivo BCG vaccination of one inbred mouse strain (BALB/c) did not result in detectable cross-reactive neutralizing antibodies against SARS-CoV-2 [28]; this latter finding again raises the possibility that genetic background influences BCG-induced phenotypes as one inbred mouse strain has no genetic diversity and can represent a limited number of humans in this regard.

Here, we also used in silico analysis to show homology between predicted antibody epitopes on the SARS-CoV-2 spike protein and the proteome of BCG. We also report the results of a pilot study to determine whether BCG vaccination of Diversity Outbred mice, and a different inbred mouse strain (C57BL/6J, not BALB/c) could induce antibodies that bind to SARS-CoV-2 spike glycoprotein. We compared the level of SARS-CoV-2 reactive antibodies in the sera of BCG-vaccinated, Diversity Outbred, and inbred C57BL/6J mice with the sera of unvaccinated Diversity Outbred mice. Sera from some, but not all, BCG-vaccinated Diversity Outbred mice contained higher levels of IgG cross-reactive to SARS-CoV-2 spike protein than sera from BCG-vaccinated C57BL/6J inbred mice and unvaccinated mice.

## 2. Materials and Methods

In Silico Analysis. BepiPred Linear Epitope Prediction 2.0 from IEBD was used to select predicted antibody binding epitopes (>5 amino acids long) on the SARS-CoV-2 spike protein (SwissProt: P0DTC2). Basic Local Alignment Search Tool (BLAST) from NCBI was used to align the selected epitopes with the Swiss-Prot reviewed proteome of BCG strain Pasteur (*Mycobacterium bovis* (strain BCG/Pasteur 1173P2)). The results were restricted to only matches with >70% ID and alignment length ≥7 amino acids, based on previous research on homologous peptide length and T cell activation [29]. Pentamers with 100% ID and 6-mer with 83.3% ID were also included, because antibody epitopes can range from 4–12 amino acids [30]. The BCG protein and cellular components, as described in the UniprotKB database, was obtained for each peptide fitting the above criteria and graphed using the R software version 4.1.0 package *ggplot2*.

BCG-vaccinated and unvaccinated mice. BCG vaccination (Figure 1 and Table 2) was performed under protocols reviewed and approved by the Institutional Animal Care and Use Committee (IACUC) of CBER/FDA protocol #2011-14. Unvaccinated mice were approved by Tufts University’s IACUC protocols G2015-33 or G2018-02, or the CEBR/FDA protocol #2011-14. Briefly, female C57BL/6J and female Diversity Outbred (Diversity Outbred) mice were obtained from The Jackson Laboratory (Bar Harbor, ME, USA) at 6–10 weeks of age. All mice were housed in groups by strain in microisolator cages and given autoclaved food and water ad libitum.

A single, frozen vial of BCG Pasteur was thawed and diluted in sterile PBS. Mice were vaccinated intradermally or intravenously with 10^5^ BCG Pasteur colony forming units as described in [31]. Eight weeks later, the mice were euthanized, and blood was obtained via cardiac puncture with a heparinized 1 mL syringe and 26 g needle. Serum was separated using Sarstedt microtubes (Fisher Scientific, Pittsburg, PA, USA) as described in [31] or allowed to clot and then centrifuged for separation. Serum samples were stored at −80 °C and shipped frozen from the Food and Drug Administration (Silver Spring, MD, USA) to the Cummings School of Veterinary Medicine (Grafton, MA, USA) for assays. All serum samples were stored at −80 °C until use.

In-house ELISA for SARS-CoV-2 Antigens. We optimized an in-house ELISA to detect cross-reactive IgG antibodies in mouse sera following methods described by Bates et al. [32]. Briefly, on day 1, Costar 96-well plates were coated in 1 μg/mL recombinant SARS-CoV-2 spike protein (“rSpike”, BEI Resources NR-52308) diluted in PBS. Half of the plate was left uncoated to measure nonspecific interactions [33]. Plates were incubated overnight at 4 °C. On day 2, plates were blocked with 1% Oxoid skim milk + sodium azide for 2 h at 25 °C then washed six times with 1× PBS-Tween. To create the standard curve, BEI Resources (NR-616) Mouse Monoclonal Anti-SARS-CoV Spike Protein Similar to 240C was serially diluted starting at 10,000 pg/mL in 1% BSA (Appendix A). As a positive control, aliquots of Mouse Monoclonal Anti-SARS-CoV Spike Protein Similar to 240C were prepared in 1% BSA at 20,000 pg/mL and frozen at −80 °C for consistency across ELISAs (Appendix A). All mouse sera were initially diluted 1:500 in 1% BSA and tested in triplicate. Negative control wells contained 1% BSA. Plates were incubated overnight at 4 °C. On day 3, plates were washed six times with 1× PBS-Tween, and horseradish peroxidase (HRP) conjugated goat anti-mouse IgG (Rockland #610-13) was used to detect SARS-CoV-2-bound IgG. Plates were washed six times with 1× PBS-Tween. Plates were incubated for 20 min with 3,3′,5,5′-Tetramethylbenzidine (TMB), and the colorimetric reaction was stopped with 0.25 M HCl. Absorbance was read at 450 nm using a BioTek plate reader (Agilent Technologies, Winooski, VT, USA). The concentration of bound-IgG was computed based on a 4-parameter logistic (4PL) regression model of the standard curve using the program Gen5.

Technical assay controls. On each plate, an independent assay control was run to assess the quality of the detection antibody and colorimetric reaction as follows: On day 1, the well was coated with 5 μg/mL of a mixture containing recombinant Early Secreted Antigenic Target-6 (ESAT-6) and Culture Filtrate Protein-10 (CFP-10) (BEI Resources NR-49424, NR-49425). ESAT-6 and CFP-10 are antigens present in virulent mycobacteria including wild type *M. bovis*, but they are not present in BCG [34]. The mycobacterial antigens and sera from immunized mice served as technical assay controls to ensure that the reagents were of good quality and worked consistently. On day 2, sera from CFP-10-immunized mice were diluted 1:100 in 1% BSA and incubated in the well. On day 3, the same detection antibody, goat anti-mouse IgG-HRP, was added, followed by TMB and then HCl, as described above.

Graphical and Statistical Analysis. The graphs were made using Prism GraphPad version 9.1.0. The limit of detection was calculated based on the sum of the mean optical density and three times the standard deviation based on 6 wells containing diluent instead of the standard. Plots and statistical analyses (ANOVA and Welch *t*-tests) were performed using R.

## 3. Results

In Silico Analysis of Homology. Potential cross-reactive antibody binding sites were predicted by in silico comparison of antibody epitopes on the SARS-CoV-2 Spike protein and the proteome of BCG. Unlike other in silico analysis, only predicted linear epitopes >5 amino acids from the spike protein receptor-binding domain (RBD) were searched against a database of Swiss-Prot reviewed BCG Pasteur proteins. The resulting homologous BCG proteins were characterized by their location on/in *M. bovis*. Bepipred Linear Epitope Prediction 2.0 from IEBD found 25 potential epitopes ≥ 5 amino acids long on the SARS-CoV-2 spike protein. A BLAST search comparing epitopes with the BCG strain Pasteur proteome yielded 186 peptides (percent ID ≥ 70% and alignment length ≥ 7 amino acids, percent ID ≥83.3% and 6-mer, percent ID 100% and 5-mer). The frequency of each percent ID, alignment length pair is described in Figure 2a. Of note, one heptamer had 100% identity with a BCG protein. As only Swiss-Prot reviewed proteins from the BCG Pasteur proteome were used in the BLAST search, all 186 peptides matched with a described BCG protein. Of those BCG proteins, the cellular component was known for 59.7%. Most BCG proteins were cytoplasmic (Figure 2b); however, five proteins are known to be located on the cell surface (Table 3). These analyses imply shared amino acid sequences between BCG surface proteins and the SARS-CoV-2 spike protein that could be epitopes recognized by antibodies.

### Concentration of Cross-Reactive IgG to SARS-CoV-2 Antigens Using ELISA

The signal from uncoated wells incubated with the sera was undetectable. The unvaccinated Diversity Outbred mice were not challenged by any pathogen, and therefore, the calculated IgG values less than 150 pg/mL may be noise, reflecting non-specific interactions between normal serum components and other assay reagents.

The concentration of IgG binding SARS-CoV-2 rSpike protein was measured in Diversity Outbred mice vaccinated with BCG, C57BL/6J mice vaccinated with BCG, and unvaccinated Diversity Outbred mice (Figure 3). Levels of cross-reactive IgG in serum samples from C57BL/6J mice vaccinated intradermally (ID) (n = 5) and intravenously (IV) (n = 5) were compared with Diversity Outbred mice vaccinated ID (n = 5), IV (n = 10), and unvaccinated (n = 5). One-way analysis of variance determined a significant difference between the means of the 5 groups (F value 4.351, *p* = 0.0083). Welch two sample *t*-tests were used to determine if the mean of each group was different from the control (unvaccinated mice). IgG concentration in BCG-vaccinated Diversity Outbred mouse sera (both ID and IV, 275 pg/mL) was significantly greater than levels in unvaccinated Diversity Outbred mice (148 pg/mL; Welch *t*-test, *p* = 0.001). In contrast, levels of cross-reactive IgG in BCG-vaccinated C57BL/6J (both ID and IV) were not significantly greater (*p* = 0.926). In two-way analysis of variance comparing BCG-vaccinated Diversity Outbred mice and BCG vaccinated C57BL/6J, mouse strain was identified a significant variable (F value 11.676, *p* = 0.002), suggesting that genetic background contributes to the variation in the levels observed. Sera from C57BL/6J mice vaccinated intradermally (ID) (n = 5, 177 pg/mL) and intravenously (IV) (n = 5, 115 pg/mL) were compared with sera from Diversity Outbred mice vaccinated ID (n = 5, 288 pg/mL) and IV (n = 10, 269 pg/mL). By Two-way analysis of variance indicated that the route of vaccination had no significance (F value 0.943, *p* = 0.342).

A subset of the BCG-vaccinated Diversity Outbred mice (10/15 or 67%) produced more antibodies reactive to the recombinant spike protein than did unvaccinated Diversity Outbred mice and BCG-vaccinated C57BL/6J mice. The mean IgG concentration reactive to the spike protein in BCG-vaccinated Diversity Outbred mice (275 pg/mL) was significantly different from the mean IgG concentration in unvaccinated Diversity Outbred mice (148 pg/mL; Welch *t*-test, *p* = 0.001). The IgG concentration in BCG-vaccinated C57BL/6J (145 pg/mL) was not significantly different from that in unvaccinated Diversity Outbred mice (*p* = 0.926). Two-way analysis of variance comparing BCG-vaccinated Diversity Outbred mice and BCG vaccinated C57BL/6J indicated that strain was significant (F value 11.676, *p* = 0.002).

The levels of cross-reactive IgG were analyzed by route of administration. Sera from C57BL/6J mice vaccinated intradermally (ID) (n = 5, 177 pg/mL) and intravenously (IV) (n = 5, 115 pg/mL) were compared with sera from Diversity Outbred mice vaccinated ID (n = 5, 288 pg/mL) and IV (n = 10, 269 pg/mL). Two-way analysis of variance indicated that the route of vaccination had no significance (F value 0.943, *p* = 0.342). Based on these results, route of vaccination has no impact on levels of cross-reactive IgG.

## 4. Discussion

Early in the COVID-19 pandemic, epidemiological evidence suggested cross-protection between the BCG vaccine and SARS-CoV-2 infection or COVID-19 disease outcomes [4], while later randomized clinical trials reported a spectrum of associations that seem to vary by country of origin [5,9,10,11,12,13,14,15,16,17]. This fueled investigation into a possible genetic influence on these responses. Using a mouse study where environment and experimental variables could be controlled, we investigated whether genetic background could influence host response to BCG vaccination in terms of the production of antibodies cross-reactive to SARS-CoV-2. We used Diversity Outbred mice as they are as genetically diverse as humans, with abundant heterozygosity, and thus may better model the human range of phenotypic responses to stimuli [35] including responses to BCG vaccination [31] than inbred strains of mice. We observed that two-thirds of BCG-vaccinated Diversity Outbred mice generated IgG cross-reactive to SARS-CoV-2 spike protein, while one-third did not; C57BL/6J inbred mice did not either, as has been observed in inbred BALB/c mice [28]. Although these studies cannot identify the causal genes that explain differences in the BCG-induced antibodies cross-reactive to SARS-CoV2 spike protein, it is clear that genetic background has an effect on BCG vaccination responses to tuberculosis [31] and on SARS-CoV-2 vaccination responses to SARV-CoV-2 [36]. Based on findings in panels of inbred mouse strains, antibody responses are heritable and are associated with specific genetic markers [37]. For example, Kelsey et al. (2020) identified 23 loci and multiple candidate genes associated with antibody responses to Influenza A virus (IAV) in Collaborative Cross inbred mice [38]. Interestingly, the loci did not include the mouse major histocompatibility locus, chromosomal locations of the T cell receptor chains, or the location of B cell receptor heavy and light immunoglobulin genes. These findings suggest that other genes and pathways regulate antibody responses. Since Collaborative Cross and Diversity Outbred mice are genetically related, we can speculate there may be shared genetic loci responsible for the variation in IgG we observed in SARS-CoV-2 cross-reactive antibodies induced by BCG vaccination of Diversity Outbred mice. Finally, if we can identify sets of genes and pathways that regulate host responses to BCG vaccination, we can use the information to predict whether an individual will generate protective (or not) responses to unrelated pathogens following BCG vaccination.

The main limitations reflect the nature of pilot studies, which involve a small number of experiments and small sample size. We focused on a single variant of SARS-CoV-2 with a defined recombinant spike protein (not mutations), quantified total cross-reactive IgG (not neutralizing antibodies), and used bioinformatics as main evidence of support for peptide homology (not direct experimentation with synthetic peptides). These are important future investigations, as spike protein mutations of variants, such as Omicron, are less well-recognized by existing neutralizing antibodies in mice and humans [39]. Further, although we included control sera at the same dilutions as non-vaccinated Diversity Outbred mice and control sera from BCG-vaccinated C57BL/6, we cannot rule out that BCG vaccination may generally increase non-specific binding in a manner that is present in Diversity Outbred mouse sera but not C57BL/6 sera. The study included a small number of mice and serum samples, which may not represent the full range of possible phenotypes in the Diversity Outbred mouse population and could erroneously inflate (or understate) the spectrum of IgG responses.

## 5. Conclusions

This pilot study to ascertain whether genetic background influences the production of BCG-induced antibodies cross-reactive to SARS-CoV-2 has limitations and raises more questions than it answers. One way in which this type of research may be informative is by providing a different perspective on the controversial evidence regarding the effect of BCG vaccine. Many studies attribute differences in vaccine responses to differences in methods, measurements, environments, or biological variables like age and gender. Few mention genetics as a driver of phenotypic variation. Diversity Outbred mice provide advantages for investigating the genetic basis of complex traits because environmental and biological variables can be controlled, and genotype is the main of variation. This enables discovery of the genes, alleles, and polymorphisms that may be responsible. Future studies with hundreds of samples will allow more rigorous genotype–phenotype association studies and analysis regarding the genetic basis of variation in BCG-induced antibodies that are cross-reactive to SARS-CoV-2.

## Figures and Tables

**Figure 1 vaccines-12-00242-f001:**
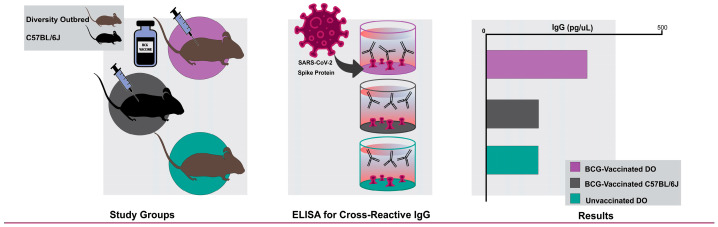
Experiment schematic. Mice were vaccinated with BCG or not vaccinated, and then IgG reactive to SARS-CoV-2 spike protein was quantified in sera by ELISA 8 weeks after vaccination.

**Figure 2 vaccines-12-00242-f002:**
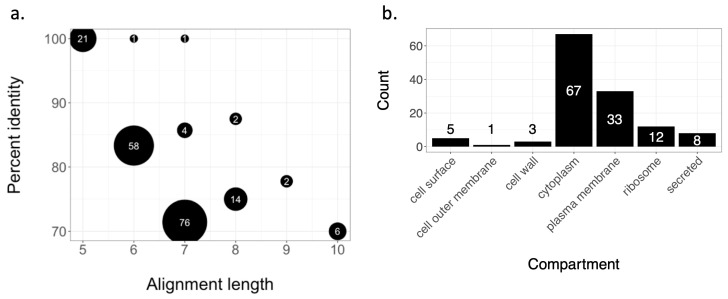
In silico analysis of SARS-CoV-2 spike protein epitopes homologous to BCG proteins. (**a**) Frequency of alignment length and percent identity pair. Peptides with homology to BCG proteins were restricted to those ≥70% identity and ≥7 amino acids. Hexamers with ≥83% identity and pentamers with 100% identity were also included. Frequency of each pair (alignment length, percent identity) is denoted by size of the data point. (**b**) Known cellular components of BCG proteins with SARS-CoV-2 homology. Peptides fitting the above-described criteria were further matched with name of the corresponding Swiss Prot *Mycobacterium bovis* strain BCG Pasteur protein. Those proteins’ cellular components were described, if known. The frequency of the cellular components was described.

**Figure 3 vaccines-12-00242-f003:**
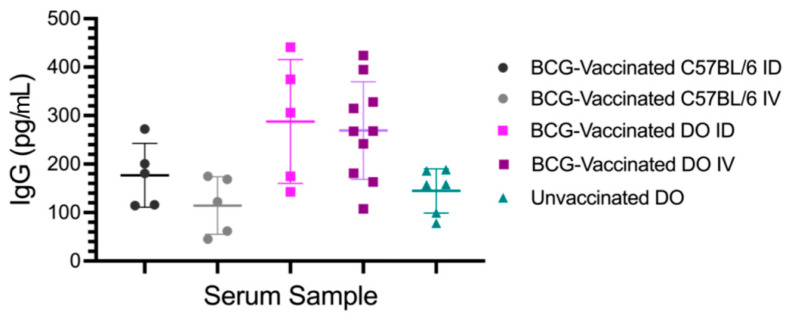
Concentrations of IgG binding SARS-CoV-2 recombinant spike protein by vaccination route and mouse strain. Each data point is the average of technical triplicates at 1:500 dilution from one mouse. ID = Intradermal; IV = Intravenous. Statistical analyses and significant differences are described in the Results section.

**Table 1 vaccines-12-00242-t001:** Summary of results from randomized clinical trials investigating effects of BCG vaccination on SARS-CoV-2 infection and COVID-19 disease.

Authors	Year	Country	Purpose	Outcomes Improved?	Immunity Enhanced?
Föhse et al. [9]	2023	Netherlands	Circadian rhythm	No	n/a
Claus et al. [10]	2023	Netherlands	Efficacy against SARS-CoV-2	No	Yes, for IgG
Pittet et al. [11]	2023	Multiple	Efficacy against COVID-19	No	n/a
Santos et al. [12]	2023	Brazil	Efficacy against SARS-CoV-2	Yes, trend	Trend for hi IgG
Faustman et al. [13]	2022	USA	Efficacy against COVID-19 in diabetes	Yes, for severity	Changed IgG score
Jalalizadeh et al. [14]	2022	Brazil	Effects in convalescent COVID-19	Yes, for anosmia	Modulated IgG
Dionato et al. [15]	2022	Brazil	Revaccination in COVID-19	Yes, for anosmia	n/a
Moorlag et al. [16]	2022	Netherlands	Efficacy against COVID-19	No	Yes, multiple
Glynn et al. [17]	2021	Malawi	Revaccination on all-cause mortality	No	n/a
Tsilika et al. [5]	2021	Greece	Efficacy against COVID-19	Yes, reduced risk	Yes, for IgG

**Table 2 vaccines-12-00242-t002:** Summary of mouse samples available for the pilot study.

Strain	Vaccination	Route of Administration	Number of Mice	Age at Euthanasia (Months)
C57BL/6J	BCG	Intradermal	5	3.5–4.5
C57BL/6J	BCG	Intravenous	5	3.5–4.5
Diversity Outbred	BCG	Intradermal	5	3.5–4.5
Diversity Outbred	BCG	Intravenous	10	3.5–4.5
Diversity Outbred	No BCG	n/a	5	5 (n = 1) 18 (n = 4)

**Table 3 vaccines-12-00242-t003:** Homologous peptides restricted to BCG cell surface proteins. The five BCG Pasteur proteins known to be located on the cell surface from Figure 2b were identified, and their sequences compared with the SARS-CoV-2 Spike protein.

Length (Pct ID)	SARS-CoV-2 Spike Protein	BCG Pasteur	Protein Name
6 (83.3%)	TSPDVD	TSADVD	Putative phthiocerol dimycocerosate transporter LppX
5 (100%)	DPSKP	DPSKP	Alanine and proline rich secreted protein Apa
6 (83.3%)	SGTNGT	SGNNGT	Phosphate binding protein PstS2
6 (83.3%)	SSGWTA	SSGGTA	Phosphate binding protein PstS2
7 (71.4%)	GQTGKIA	GQT_ _ IA	Phosphate binding protein PstS3

## Data Availability

The original datasets generated and analyzed for this study can be found at Zenodo.org, DATASET. Specht AG, Kurtz SL, Elkins KL, Specht H, Beamer, G. Data sets and analysis for BCG vaccination of Diversity Outbred mice induces antibodies cross-reactive to SARS-CoV-2 spike protein. Zenodo. 2023. doi: 10.5281/zenodo.7735191.

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
