# Peer review of "Host Genetic Background Influences BCG-Induced Antibodies Cross-Reactive to SARS-CoV-2 Spike Protein"

_vaccines, 2024, doi:10.3390/vaccines12030242_

Round 1

Reviewer 1 Report

Comments and Suggestions for Authors

This is an interesting and provocative paper which helps explain previously discrepant results in human clinical trials. The authors appropriately recognize and discuss the limitations of this preliminary study and provide suggestions for further research to follow. A few relatively minor points could be improved :

1. The total IgG levels in the different groups of mice (in Table 2) should be determined and presented. Similarity of the mean and/or median total IgG levels would provide confidence that the apparent results are not due to non-specific binding of IgG or immune complexes in immunized mice. Alternatively, if the total IgG levels are quite distinct, and particularly, if total IgG levels in BCG-immunized mice are much higher than in non-immunized mice, controls should be run with different dilutions of sera from non-immunized mice of each type studied to provide asssurance that the apparent results do not represent non-specific binding due to generally increased IgG levels in immunized mice. (See lines 196- 199). 

2. A few abbreviations should be defined and some clarifications should be added to the text: -line 150-please briefly identify ESAT-6 and CFP-10 as M. bovis antigens, with a reference.

-please define RBD in line 165

- is the term pg/mL/mL in line 198 a typo ? Please correct or explain. 

Reviewer 2 Report

Comments and Suggestions for Authors

Review of "Host genetic background influences BCG-induced antibodies cross-reactive to SARS-CoV-2 spike protein"

The authors of this study indicate that this is a report of a pilot study intending to study the effect of BCG vaccination of outbred mice to investigate the role of diverse genetic backgrounds on the antibody response to COVID-2 spike protein.  They performed this study to investigate the conflicting reports in the recent literature that some studies reported protective effects of BCG immunization in human populations and other reports indicated no such protective effects. One obvious defect in the current study (as pointed out by the investigators) pertains to the limited number of mice in each experimental group (an n of 10 or 5).  In fact, it is remarkable that such an under-powered study was able to achieve any significant results!  I would have to agree that the results do suggest that the genetic background in mice may appear to influence the development of cross-reactive antibodies to BCG and COVID-19 antigens.  However, the authors do not present any evidence that predicted homologous peptides between BCG and spike protein do in fact cross-react.   This would be a simple thing to test experimentally with synthetic peptides.  

The last issue I would like addressed in the Discussion would be where the investigators feel their work is going.  How is knowing that diversity in genetic backgrounds (mouse or human) has a strong influence on the particular epitopes that B cells respond to?  Don't we already know this based on MHC diversity?  How is your work going to improve/inform human immune responses to COVID-19?   

Round 2

Reviewer 1 Report

Comments and Suggestions for Authors

The revisions are fine and have clarified the issues in question. The revised manuscript is improved, very informative and readily understandable.